# Dose- and Segment-Dependent Disturbance of Rat Gut by Ionizing Radiation: Impact of Tight Junction Proteins

**DOI:** 10.3390/ijms24021753

**Published:** 2023-01-16

**Authors:** Alexandra A. Livanova, Arina A. Fedorova, Alexander V. Zavirsky, Igor I. Krivoi, Alexander G. Markov

**Affiliations:** 1Department of General Physiology, St. Petersburg State University, 199034 St. Petersburg, Russia; 2Department of Military Toxicology and Radiation Defense, S. M. Kirov Military Medical Academy, 194044 St. Petersburg, Russia

**Keywords:** ionizing radiation, intestine, colon, gut permeability, tight junctions, claudins, gastrointestinal acute radiation syndrome

## Abstract

The damaging effect of ionizing radiation (IR) exposure results in the disturbance of the gut natural barrier, followed by the development of severe gastrointestinal injury. However, the dose and application segment are known to determine the effects of IR. In this study, we demonstrated the dose- and segment-specificity of tight junction (TJ) alteration in IR-induced gastrointestinal injury in rats. Male Wistar rats were subjected to a total-body X-ray irradiation at doses of 2 or 10 Gy. Isolated jejunum and colon segments were tested in an Ussing chamber 72 h after exposure. In the jejunum, 10-Gy IR dramatically altered transepithelial resistance, short-circuit current and permeability for sodium fluorescein. These changes were accompanied by severe disturbance of histological structure and total rearrangement of TJ content (increased content of claudin-1, -2, -3 and -4; multidirectional changes in tricellulin and occludin). In the colon of 10-Gy irradiated rats, lesions of barrier and transport functions were less pronounced, with only claudin-2 and -4 altered among TJ proteins. The 2-Gy IR did not change electrophysiological characteristics or permeability in the colon or jejunum, although slight alterations in jejunum histology were noted, emphasized with claudin-3 increase. Considering that TJ proteins are critical for maintaining epithelial barrier integrity, these findings may have implications for countermeasures in gastrointestinal acute radiation injury.

## 1. Introduction

Ionizing radiation (IR) has a powerful damaging effect in living organisms, causing multiple organ failure [1]. Irradiation at high doses (6–15 Gy) causes severe intestinal dysfunction that characterizes gastrointestinal acute radiation syndrome (GIARS) [2,3]. People can develop gastrointestinal radiation injury as a result of whole-body exposure in radiation accidents, as well as during abdominal radiotherapy for certain types of cancer [4,5,6]. The pathophysiological basis for GIARS manifestation is a disturbance of the natural epithelial tissue barrier, reflecting in the disintegration of epithelial cells with subsequent “denudation” of the mucous membrane [3,7,8]. Moreover, an increased caspase-3 activation signal, suggesting apoptosis stimulation in intestinal crypts, is also involved in GIARS pathology [9]. The manifestation of GIARS in mammals is associated with weight loss, decreased nutrient absorption, profuse diarrhea and progressive dehydration, as well as bacterial translocation into the bloodstream [2,10].

Tight junctions (TJs), apical complexes connecting neighboring cells in barrier tissues, are known to provide the mechanical binding of enterocytes into a single layer, as well as paracellular transport control [11,12,13]. The key molecular components of TJs are proteins of the claudin family, which consists of 27 members in mammals, and the TJ-associated MARVEL-domain-containing protein (TAMP) family, primarily including occludin and tricellulin [14,15]. Multiple studies have confirmed tissue-specific heterogeneity in the normal distribution of claudins and TAMP proteins along the segments of the gut, establishing a gradient of barrier and transport functions. “Sealing” claudins-1, -3 and -4 are predominantly detectable in the colon, and the expression of claudins that mediate ion permeability, mainly claudin-2, is pronounced in the jejunum and ileum of rats [16,17].

The mechanical integrity of TJs is altered under different physiological and pathological conditions. Thus, claudin dysregulation that occurs in enterocytes contributes to epithelial permeation disorder and multiple intestinal diseases, including inflammatory bowel diseases [18]. There is also growing evidence that TJ proteins may act as signaling proteins and participate in inflammation, cell proliferation, differentiation and tumorigenesis via various cellular signaling pathways [19]. The content of some TJ proteins was reported to be altered under IR exposure. Thus, multidirectional changes in the levels of claudin-2, -3 and -4 in the ileum of rats were found on the sixth day after abdominal exposure to IR at a dose of 12 Gy [20]. After total body irradiation of mice at a dose of 4 Gy, a redistribution of occludin, claudin-3 and actin cytoskeleton in the tissues of the ileum and colon was observed, indicating a high sensitivity of this system to IR [21]. Given a few reports on the role of particular claudins in the pathogenesis of gastrointestinal radiation injury, data on the dose- and segment-sensitivity of gut TJ complexes remain scarce. 

In this study, we address the dose effect and segment-specificity of TJ protein impact in IR-induced gastrointestinal injury in rats.

## 2. Results

### 2.1. High Dose IR Exposure Affects Weight Gain and Leukocyte Counts in Rats

Control rats maintained weight during the study period, and 2-Gy irradiation did not affect weight dynamics (Figure 1a). The 10-Gy-irradiated animals continuously lost weight after exposure, losing up to 16% of body weight by the third day (Figure 1a). In this group, the animals responded with profuse diarrhea, temperature decrease, hair loss and nasal hemorrhage, as previously reported [22,23,24]. 

To verify the absorbed dose, hematological assay was carried out, with the total leukocyte count and lymphocyte and neutrophil counts used as bioindicators (Figure 1b). The total leukocyte count in control animals was 8.3 ± 1.1 × 10^3^/µL 72 h after sham irradiation, which corresponds to normal reference levels for Wistar rats [25]. The exposure to 10-Gy IR significantly (*p* < 0.01) decreased the level of leukocytes in heart blood 72 h after irradiation (Figure 1b). The decrease in the level of leukocytes in this group was contributed to by lymphocytes and segmented neutrophils. The severe radiation leukopenia observed corresponds to the expected hematological pattern commonly seen in rats when irradiated at such a dose [25,26]. 

### 2.2. High-Dose IR Disturbs Transepithelial Resistence (TER) and Short-Circuit Current (Isc) of Jejunum and Colon

Transepithelial resistance (TER) and short-circuit current (*Isc*) were evaluated in an Ussing chamber as a physiological indicator of the barrier and transport functions of jejunum and colon tissues. The 2-Gy IR did not affect the electrophysiological parameters of the jejunum or colon (Figure 2a–d). Previously, while studying dose- and time-dependence of IR effects on rat intestines, 72 h after 10-Gy IR was shown to be the experimental point at which the disturbance of the barrier and transport functions of the jejunum were obtained [27]. Here, as expected, exposure to 10-Gy IR significantly and dramatically decreased TER (*p* < 0.001) and increased *Isc* (*p* < 0.01) in the jejunum and colon (Figure 2a,c). Paracellular flux of sodium fluorescein, often used as biomarker of epithelium permeability [28,29], was also increased in the jejunum (*p* < 0.01) in the 10-Gy-irradiated group (Figure 2e).

In the colon, electrophysiological characteristics were also disturbed in 10-Gy-irradiated animals (Figure 2b,d). Paracellular flux of sodium fluorescein in the colon did not change significantly (Figure 2f).

Alteration in electrophysiological characteristics suggests a lesion of the barrier properties (TER) and transport across the epithelium (*Isc*) after IR exposure at high doses. At the same time, TER in the jejunum was altered more dramatically compared to the colon (79% decrease compared to control in jejunum, and 36% in colon). The less significant impairment of TER in the colon, as well as unchanged permeability, indicates increased colonic barrier radioresistance compared to the jejunum, as previously confirmed [30,31].

### 2.3. IR in Low and High Doses Disturbs Histological Structure of Jejunum and Colon Tissue

Hematoxilin and eosin (H&E)-stained sections of the jejunum and colon from sham-irradiated control animals showed normal morphological structure (Figure 3a). The crypt-villus architecture in the jejunum was damaged at 10-Gy IR, with villi being shortened and thickened, and crypt depth and width being decreased (Figure 3a,b). In the colon, 10-Gy IR caused significant crypt depth decrease (*p* < 0.01) (Figure 3c,d). The 2-Gy irradiation did not alter villus height in the jejunum, although it caused a significant increase in villus width (*p* < 0.01), crypt depth (*p* < 0.05) and crypt width (*p* < 0.05) (Figure 3a,b). IR-induced damage to the gut villus–crypt axis reflects a disturbance in the balance between epithelial cell proliferation and apoptosis. In the jejunum, wall thickness was increased significantly in the 10-Gy-irradiated group (*p* < 0.001), mainly due to the submucosal layer (*p* < 0.001), whereas muscular and adventitia layers were not altered. In the colon, wall thickness was increased significantly both in the 2-Gy- and 10-Gy-irradiated groups (*p* < 0.01 and *p* < 0.001, respectively), mainly due to muscular and adventitia layers (Figure 3c,d). 

### 2.4. IR Disturbs Tight Junction (TJ) Proteins in Jejunum and Colon in Dose- and Segment-Specific Manners

TJ protein content was measured in homogenates of the jejunum and colon sections after rat exposure to IR. Exposure to 10-Gy IR significantly increased the protein content of claudin-1, -2, -3, -4 and occludin in the jejunum, and decreased the level of tricellulin (*p* < 0.05 each) (Figure 4a,b). The 2-Gy IR significantly increased the level of claudin-3 in the jejunum only (*p* < 0.05). The level of 17 kDa cleaved caspase-3, used as an indicator of apoptosis [9,10,32], did not change after 2-Gy exposure, although it was significantly decreased in the jejunum after 10-Gy irradiation (*p* < 0.05) (Figure 4a,b).

In the colon, the levels of claudin-2 and claudin-4 significantly increased when exposed to 10-Gy IR (Figure 4c,d). When exposed to radiation at a dose of 2 Gy in the colon, none of the studied proteins changed.

## 3. Discussion

The normal functioning of TJ proteins is vital for the mechanical integrity of barrier tissue, effective separation of the organism from the environment, and maintenance of homeostasis. A defective intestinal TJ barrier is a contributing factor to pathogenic conditions of the gut, including celiac disease, inflammatory bowel disease and necrotizing enterocolitis [33,34]. Increased intestinal permeability caused by variations in TJ proteins was shown to result in bacterial translocation [35]. At the same time, barrier disturbance, bloodstream contamination with bacteria and subsequent endotoxemia are the main drivers of lethal outcome in the development of GIARS after irradiation exposure [36]. Could TJ disturbance be involved in the molecular mechanism of gastrointestinal irradiation injury? Previously, intestinal TJ proteins, particularly claudins, were shown to be altered under IR exposure in mammals [20,21]; however, data on the dose-dependence and segment-sensitivity remain scarce.

Previously, we examined TER, *Isc*, paracellular permeability of sodium fluorescein and the histological structure of the jejunum under 2-, 5- or 10-Gy exposure, considering 72 h as an optimal interval to assess irradiation effects in Wistar rats [27]. Here, we used a 72 h post-irradiation time point and compared 2-Gy and 10-Gy doses, which are above and below the established LD50/30 in this animal model [37]. In this study, we found that the negative effect of IR on the gastrointestinal tract of rats is dose- and segment-dependent, and this diversity is emphasized by various patterns of TJ alterations.

In general, the segment-specific gradient of the barrier properties of the intestine, both under normal and pathological conditions, coincides with the mosaic of TJ proteins. For instance, in the duodenum of rats, an increased TER was previously recorded compared to the jejunum, reflecting different molecular patterns of TJ proteins detected in these sections (claudin -2, -3, -4, -5, -8 in the duodenum, and claudin -1, -2, -7, -12 in the jejunum) [13]. The epithelium of Peyer’s patches was characterized by higher TER compared to the villous epithelium, and the content of cludins-1, -4, -5, -8 increased [38]. In this study, a dramatic disturbance in the barrier and transport functions was observed in rat jejunum after IR exposure, which is consistent with the idea of the higher radiosensitivity of the small intestine compared to the colon [7]. We also demonstrate different patterns of TJ proteins presented in the jejunum (claudin-1, -2, occludin and tricellulin) and colon (claudin-1, -2, -3, -4, occludin and tricellulin) of rats. In addition, it was found that when exposed to IR, a pronounced alteration in TJ content occurs in the jejunum (increased levels of claudin-1, -2, -3, -4, multidirectional changes in tricellulin and occludin), whereas in the colon, only the levels of claudin-2 and claudin-4 significantly changed. These results demonstrate a segment-specific IR effect and illustrate that a less severe colonic radiation response is accompanied by less dramatic disturbance in the TJ complexes in this section.

Claudins are conventionally divided into those capable of sealing, reducing paracellular permeability and pore-forming, capable of transport stimulation throughout the epithelium [39]. Changes in the content of claudins of both types previously accompanied alterations in the permeability of the intestinal epithelium in various pathological conditions. Thus, an increase in the expression of sealing claudin-1 and pore-forming claudin-2 was observed in mammals during inflammation [40,41,42,43]. In this article, we also demonstrated an increase of claudin-1 and -2 in the jejunum after exposure to IR at a dose of 10 Gy. At the same time, it was claudin-2 which significantly increased both in the jejunum and in the colon after IR exposure. Claudin-2, being a pore-forming TJ protein, could influence the acceleration of water transport and the development of profuse diarrhea in the 10-Gy-irradiated group. In addition, claudin-2 is known to suppress the immune response by inhibiting pro-inflammatory signaling of NF-ƙB and STAT-3 through TGF-β synthesis activation [44]. Here, an increase in claudin-2 72 h after IR exposure might be additionally considered in the context of the anti-inflammatory response. 

Unlike claudin-1, which is spread among the gut segments, claudins-3 and -4, also classified as sealing proteins, are not found in the jejunum of Wistar rats normally [17]. At the same time, both claudin-3 and -4 were found in the jejunum of 10-Gy-irradiated rats. Claudin-3 was previously considered as a potential marker for IR-induced intestinal barrier failure [20]. Surprisingly, we observed a claudin-3 increase in the jejunum of 10-Gy-irradiated rats 72 h after exposure instead of a decrease, reported previously in rats 6 days after exposure [20]. This can be explained by the fact that claudin-3 is not detected in the intact jejunum of Wistar rats, and the mechanisms of its participation in the radiation reaction can be different compared to other gut segments investigated. Furthermore, there are conflicting data on the non-canonical functions of claudin-3. Thus, claudin-3 is known as *Clostridium perfringens* enterotoxin receptor; their interaction induces pore formation in the plasma membrane of the host mucosa cells and rapid cytolysis [45]. Claudin-3 was also shown to induce cancer stemness via estrogen receptor-α [46], thus acting as a proliferation activator. Notably, claudin-3 was the only protein whose content was triggered by ionizing radiation at a dose of 2 Gy in our study. This may indicate a key role of this protein in the regulation of intestinal IR response, even upon exposure at low dose, although the exact mechanism of claudin-3 interactions remains unclear.

Claudin-4 was also revealed in the jejunum of rats in the 10-Gy-irradiated group, as well as in the colon, where it was increased compared to control. Previously, it was shown that, in addition to the canonical sealing role in TJ complexes, claudin-4 regulates the apoptotic response and proliferation rate through wide signaling networks [47,48]. The loss of claudin-4 expression significantly increased caspase-3 activation and reduced tumor cell migration [49]. Here, overexpressed claudin-4 in the jejunum and colon, coinciding with a caspase-3 decrease 72 h after 10-Gy IR exposure, can reflect the role of claudin-4 in apoptosis inhibition and proliferation stimulation. During GIARS, radiation-induced apoptosis leads to interrupted migration of epithelial cells from the crypts to the tips of the villi, followed by denudation of the intestinal mucosal barrier.

Free radical formation with reactive oxygen species (ROS) generation is also described as a possible mechanism of cell damage under IR exposure, as free radical scavengers have shown success in the mitigation of GI-ARS [10,50]. Occludin, as a protein important for a stable TJ strand network, was shown to be triggered by ROS-induced tyrosine phosphorylation and redox-sensitive dimerization [51,52]. An increase in occludin expression has already been recorded in the bladder epithelium after IR exposure [53], which is consistent with our results. Notably, induced overexpression of small intestine occludin previously suppressed tumor growth by modulating a set of several apoptosis-associated genes [54]. Perhaps, within the framework of IR-induced mitotic catastrophe, an increase in occludin content is a link to the triggering of apoptosis. This is also supported by proteomic data demonstrating a shared protein environment for claudin-4 and occludin, which are involved in apoptotic response and proliferation rate regulation [47].

Tricellulin is mainly localized in the areas of association of three adjacent cells: the so-called tricellular tight junctions (tTJs). Tricellulin is believed to be involved in the organization of the spatial architecture of TJs, providing their cis-interactions with proteins of other TJs [55]. The physiological role of tricellulin in bicellular TJs is to maintain the integrity of the epithelial layer, reducing strand discontinuities and decreasing paracellular permeability [56]. Tricellulin knockdown in mice resulted in the destruction of the TJ structure, both tricellular and bicellular, and was also accompanied by occludin accumulation [14]. In this study, the dramatic decrease in TER and increase in jejunal permeability to sodium fluorescein in the 10-Gy-irradiated group coincides with decrease in tricellulin levels. Multidirectional changes in tricellulin and occludin may indicate the replacement of tricellulin by occludin 72 h after IR exposure. 

Various approaches are implicated to reduce damaging effects of IR as countermeasures for GI-ARS. Mitigation of ROS via free radical scavengers, such as N-acetyl-cysteine (NAC), have shown improvement in malondialdehyde and glutathione levels, as well as caspase-3 expression in the small intestine of rats [10]. Some studies highlight the role of TJ proteins in gut protection with drug candidates. Neurotensin, a gut tridecapeptide that stimulates the growth of normal gut mucosa and pancreas, was shown to significantly reduce bacterial translocation and restore the structure of the villi caused by IR exposure, with subsequent influence on claudin-3 level in mice [20]. In our previous study, chronic ouabain administration was proven to modulate the transport and barrier functions of rat colon epithelium damaged by IR exposure [57]. There is growing evidence that ouabain in nanomolar concentrations can affect the expression of claudins and epithelium barrier properties by triggering cSrc/Erk1/2 intracellular signal pathways and Na,K-ATPase [29,58], which makes it a potential candidate to protect the gut from ionizing radiation in the context of TJ proteins.

The results of this study demonstrate that when exposed to IR, the barrier and transport functions of the jejunum and colon of rats change, and the severity of these changes is dose- and segment-specific. Functional damage to the intestine under irradiation is accompanied by multidirectional changes in the level of TJ proteins. Alterations of TJ complexes may underlie not only damage to the intestinal barrier, but also other pathophysiological processes that occur during the development of gastrointestinal radiation injury, participating in inflammation, apoptosis and proliferation rate regulation. Given the wide functional range of TJ proteins, our findings may have wide implications for countermeasures in gastrointestinal acute radiation injury.

## 4. Materials and Methods

### 4.1. Animals

Male Wistar rats *(Rattus norvegicus*) (200–220 g, n = 20) were used in the experiments. Animals were housed in a temperature- and humidity-controlled room with food and water ad libitum. A 12 h light/dark cycle was maintained, with a constant air temperature of 22 ± 2 °C. All procedures involving rats were performed in accordance with the recommendations for the Guide for the Care and Use of Laboratory Animals [59]. The experimental protocol was approved by the Ethical Committee for Animal Research at St. Petersburg State University No. 131-03-5 dated 13 December 2017, and complied with the EU requirements set forth in Directive 2010/63/EU on animal experiments.

Animals were randomly subdivided into three groups. The rats of the two experimental groups were subjected to a single total external exposure of X-ray radiation at doses of 2 Gy (n = 4) or 10 Gy (n = 6) using the RUM-17 orthovoltage therapeutic X-ray unit (MosRentgen, Russia). During irradiation, the animals were placed in an aerated plexiglass box, which completely restricted their movement. The focal length of the X-ray tube was 50 cm; dose rate—0.31 Gy/min. To check the absorbed dose, an individual dosimeter was used, followed by result interpretation with a GO-32 measuring device (Spetsoborona, Russia). The control group of rats (n = 10) was subjected to a procedure of sham irradiation, in which the animals were placed in a box under the deactivated X-ray tube for 30 min. Rat body weight was measured at the beginning of protocol and daily during the experiment.

At 72 h after irradiation, the animals were subjected to deep anesthesia by administration of tribromoethanol at a dose of 750 mg/kg of body weight. After the autopsy procedure, freshly isolated jejunum and colon fragments were immediately used for electrophysiological measurements. Blood was taken from the heart to provide hematological analysis, and other fragments of the jejunum and colon were collected and then stored either at −80 °C for later Western blot analysis, or in 10% formalin for histological assessment. Animals were sacrificed by cervical dislocation.

### 4.2. Hematological Analysis

Heart blood samples were collected in sterile vacuum tubes containing EDTA-K3 (Chengdu Puth Medical Plastics Packaging Co., Chengdu, China) for hematological studies. The total leukocyte count in the blood was determined using an XN-1000 automatic hematology analyzer (Sysmex Corporation, Kobe, Japan). The number of leukocytes was expressed as the number of cells per 1 μL of blood. To count individual leukocyte fractions (lymphocytes and segmented neutrophils), thin blood smears were prepared, stained with azure II-eosin according to Romanovsky–Giemsa (see [60]). The resulting smears were analyzed using a Leica DMI6000 light microscope (Leica, Wetzlar, Germany); the relative content of lymphocytes and segmented neutrophils was calculated per 100 detected leukocytes and expressed as the number of cells per 1 μL of blood.

### 4.3. Registration of Electrophysiological Parameters in the Ussing Chamber

Samples of jejunum and colon were mounted in the Ussing chamber to register electrophysiological parameters according to the method described previously [27,61]. The solution used in the experiments contained (in mM): NaCl, 119; KCl, 5; CaCl_2_, 1.2; MgCl_2_, 1.2; NaHCO_3_, 25; Na_2_HPO_4_, 1.6; NaH_2_PO_4_, 0.4; D-glucose, 10 (pH 7.4). During registration (60 min), the temperature of 37 °C was maintained in the chambers. The solution filling the glass tanks of each chamber was continuously gassed with carbogen (95% O_2_ and 5% CO_2_) during the recording.

The value of the short-circuit current (*Isc*) was recorded when the voltage was fixed at the zero level (0 mV). To determine the transepithelial resistance (TER) of the tissue, the voltage was recorded when the current was fixed at a value of 10 μA. Transepithelial resistance was calculated according to Ohm’s law, taking into account the area of the camera aperture (0.126 cm^2^), and expressed in Ω × cm^2^.

### 4.4. Permeability for Sodium Fluorescein

To measure the permeability of the jejunum and colon, sodium fluorescein solution (Sigma Aldrich, Darmstadt, Germany) was placed into the Ussing chamber from the apical side to obtain a final concentration of 0.1 mM. The solution from the basolateral side was collected after 60 min of incubation to determine the concentration of sodium fluorescein permeated through the tissue. The signal intensity was measured in 96-well plates using a Typhoon FLA 9500 laser scanner (GE, Piscataway, NJ, USA), with an excitation wave length of 473 nm and a voltage of 430 V. The obtained image of the plate was analyzed using ImageJ software (NIH, Madison, WI, USA). The permeability value (Papp) was calculated using the formula Papp = (dQ/qt)/(A × C_0_), where (dQ/qt) is the concentration of sodium fluorescein on the serous side after 60 min of incubation (mol/L); A is the area of the studied tissue area (cm^2^); and C_0_ is the concentration of sodium fluorescein in the solution from the mucosal side at the initial time (mol/L).

### 4.5. Histological Analysis

Segments of the jejunum and large intestine obtained from sham-irradiated animals (n = 3), as well as from animals irradiated at a dose of 2 Gy (n = 3) and 10 Gy (n = 3), were fixed in 10% formalin solution (BioVitrum, Saint-Petersburg, Russia), followed by paraffin embedding. Using a Leica RM2265 rotary microtome (Leica, Wetzlar, Germany), 5 µm tissue sections were obtained, mounted on SuperFrost glass slides (Thermo Fisher Scientific, Waltham, MA, USA) and stained with hematoxylin and eosin (H&E). The obtained histological sections were analyzed using a Leica DMI6000 light microscope (Leica, Wetzlar, Germany). The images were obtained using a color digital CCD camera (Leica, Wetzlar, Germany) at a magnification of ×100 and ×200 using the Leica Application Suite software (Leica, Wetzlar, Germany).

To assess the changes in the histological structure of the tissues of the jejunum and colon after irradiation, quantitative analysis of the obtained images was performed using ImageJ software (NIH, Madison, WI, USA). In each animal, two tissue fragments were obtained, for each of which there were at least two H&E-stained sections; at least five visual fields were examined per each section. Data on every five visual fields, averaged per each H&E-stained section, were taken as an independent event. Independent double-blind analysis was conducted while providing quantitative assessment of tissue histological structure. The following morphometric parameters were evaluated in the jejunum: villus height and width, crypt depth and width, wall thickness, submucosa layer thickness, muscle and adventitia layer thickness; in the colon: crypt depth and width, wall thickness, submucosa layer thickness, muscle and adventitia layer thickness. 

### 4.6. Western Blot

Pre-frozen (at −80 °C) tissue fragments of the jejunum and colon were incubated with RIPA buffer (10 mM Tris-HCl buffer (pH 7.4), 150 mM NaCl, 0.5% Triton X-100, 0.1% SDS), containing protease inhibitors Complete mini-EDTA-free tablets (Roche, Basel, Switzerland). Mechanical homogenization was carried out using a Retsch MM400 homogenizer (Retsch, Haan, Germany) followed by centrifugation for 15 min at 13,000× *g* at 4 °C (Eppendorf, Hamburg, Germany). 

Total protein contents in the supernatants were measured with a Pierce Rapid Gold Bicinchoninic Acid Protein Assay Kit (Thermo Fisher Scientific, Waltham, MA, USA), according to the manufacturer’s protocol, using a spectrophotometric microplate reader SPECTROstar Nano (BMG Labtech, Ortenberg, Germany). Equal amounts of total protein were heated for 10 min at 95 °C with a 4× Laemmli buffer.

Electrophoresis of protein samples was performed in 10% polyacrylamide TGX Stain-Free™ FastCast™ gel (Bio Rad, Hercules, CA, USA). Proteins were transferred from the gel onto polyvinylidene fluoride (PVDF) membranes (Bio Rad, Hercules, CA, USA) preactivated in methanol (30 s; 24 °C) using the Trans-Blot Turbo Transfer System (Bio Rad, Hercules, CA, USA).

After blocking in 5% milk powder, PVDF membranes were incubated overnight with primary mouse antibodies for claudin-2, and occludin or primary rabbit antibodies for claudin-1, -3, -4, tricellulin (#32-5600, #33-1500, #71-7800, #34-1700, #36-45800, #48-8400, Thermo Fisher Scientific, Waltham, MA, USA) and cleaved caspase-3 (#9661s, Cell Signalling, Danvers, MA, USA). After washing, PVDF membranes were incubated for 45 min in a solution of secondary goat anti-mouse or goat anti-rabbit antibodies conjugated with horseradish peroxidase (#AB205719, #AB205718, Abcam, Cambridge, UK). For chemiluminescent detection of proteins of interest, membranes were incubated in Clarity™ Western ECL solution (Bio-Rad, Hercules, CA, USA). The signal from bound secondary antibodies was detected using the ChemiDoc XRS+ Imaging System (Bio-Rad, Hercules, CA, USA). Band intensities were normalized using Image Lab 6.1 Software (Bio-Rad, Hercules, CA, USA) to the total protein load in the same sample measured in the membrane prior to incubation with antibody. 

### 4.7. Statistics

Statistical analysis was performed using GraphPad Prism 8 software (GraphPad, San Diego, CA, USA). The difference between groups was estimated using two-way or one-way ANOVA followed by Bonferroni multiple comparisons test. When comparing the results of hematological analysis, the nonparametric Mann–Whitney test was used to compare 10-Gy irradiated group with control. All data are presented as mean ± standard error of the mean.

## Figures and Tables

**Figure 1 ijms-24-01753-f001:**
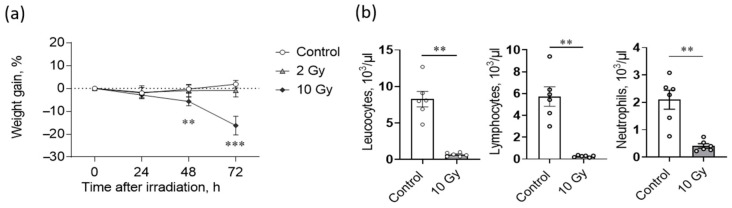
The effects of ionizing radiation (IR) on weight gain (**a**) and leukocyte counts (**b**) of rats. (**a**) Rats exposed to 10-Gy IR lost weight daily, whereas control and 2-Gy-irradiated groups did not differ significantly. Two-way ANOVA followed by Bonferroni multiple comparisons test, ** *p* < 0.01, *** *p* < 0.001—10-Gy-irradiated group vs. control. (**b**) Total leukocyte count, lymphocyte and neutrophil counts in the heart blood of rats decreased significantly after 10-Gy irradiation. The number of rats studied corresponds to the number of symbols. Mann–Whitney test, ** *p* < 0.01—10-Gy-irradiated group vs. control.

**Figure 2 ijms-24-01753-f002:**
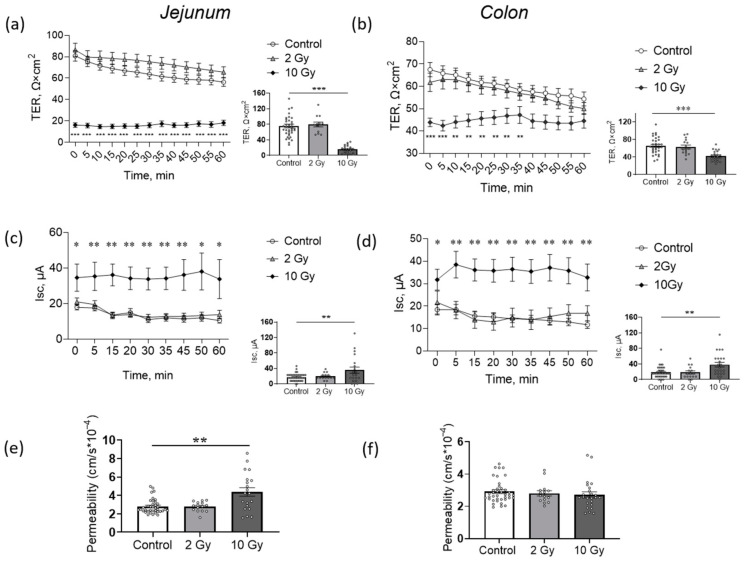
Transepithelial resistance (TER), short-circuit current (*Isc*) and permeability of jejunum (**a**,**c**,**e**) and colon (**b**,**d**,**f**) of rats after IR exposure. (**a**,**b**) TER dynamics during 60 min registration in Ussing chamber (line charts, left) and values, measured at 5 min time point (box plots, right). Two-way and one-way ANOVA followed by Bonferroni multiple comparisons test, ** *p* < 0.01, *** *p* < 0.001—10-Gy-irradiated group vs. control. (**c**,**d**) Short-circuit current (*Isc*) dynamics during 60 min registration in Ussing chamber (line charts, left) and values, measured at 5 min time point (box plots, right). Two-way and one-way ANOVA followed by Bonferroni multiple comparisons test, * *p* < 0.05, ** *p* < 0.01—10-Gy-irradiated group vs. control. (**e**,**f**) Jejunum and colon permeability measured as the paracellular flux of sodium fluorescein. Number of symbols corresponds to tissue segments in Ussing chamber; for each rat, 3–4 segments of tissue were examined. One-way ANOVA followed by Bonferroni multiple comparisons test, ** *p* < 0.01—10-Gy-irradiated group vs. control.

**Figure 3 ijms-24-01753-f003:**
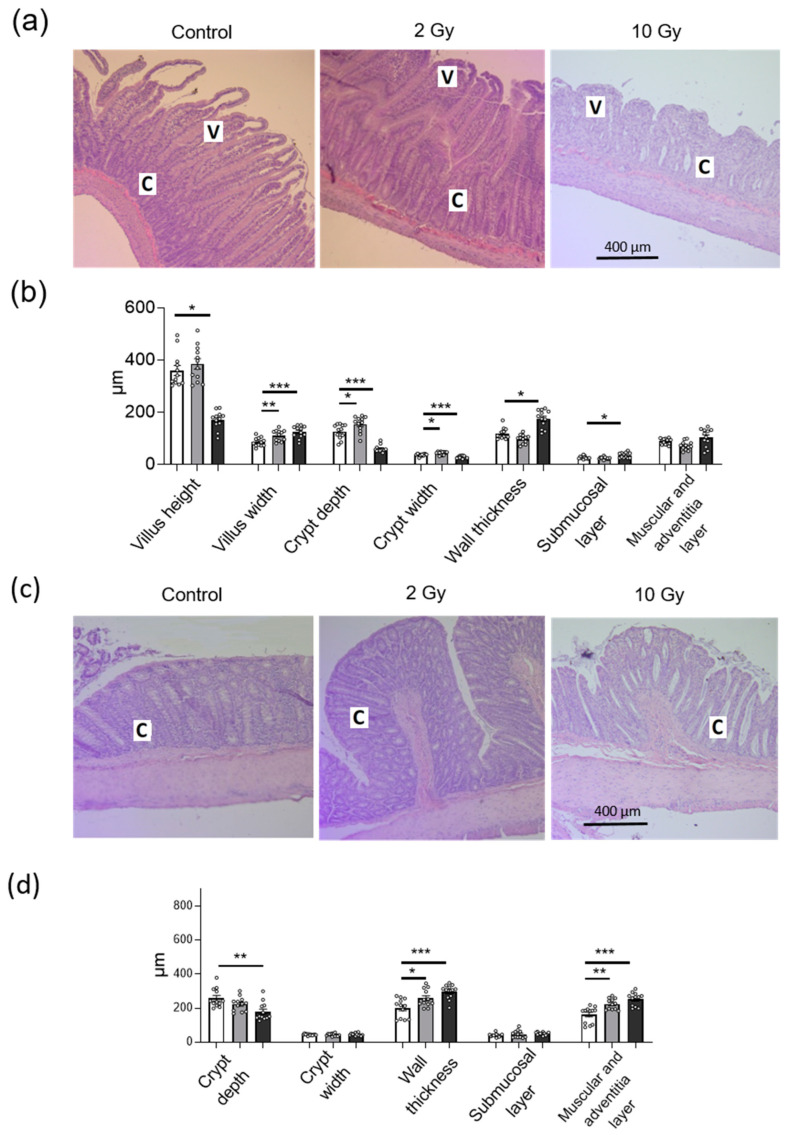
Histopathology of the jejunum (**a**,**b**) and colon (**c**,**d**) rat tissues after IR exposure. (**a**,**c**) H&E-stained sections of jejunum and colon after IR exposure; C—crypt, V—villus. (**b**,**d**) Histopathological parameters of rat jejunum and colon after IR exposure. The number of symbols corresponds to the number of H&E sections analyzed (see Section 4). One-way ANOVA followed by Bonferroni multiple comparisons test. * *p* < 0.05, ** *p* < 0.01, *******
*p* < 0.001—2-Gy- or 10-Gy-irradiated group vs. control.

**Figure 4 ijms-24-01753-f004:**
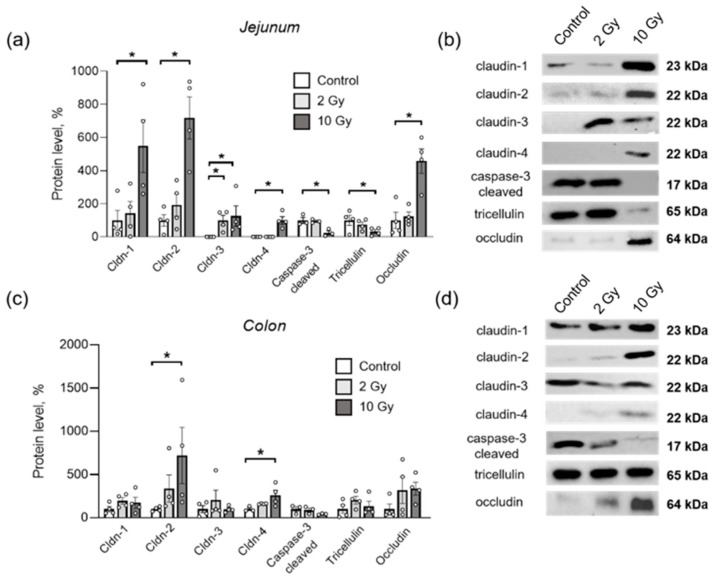
Content of TJ proteins and cleaved caspase-3 in rat jejunum (**a**,**b**) and colon (**c**,**d**) after IR exposure. (**a**,**c**) Stain-Free™ Western Blot analysis of protein level. Number of symbols corresponds to the number of rats studied. Band intensities were normalized using the total protein load. One-way ANOVA followed by Bonferroni multiple comparisons test. * *p* < 0.05—2-Gy- or 10-Gy-irradiated group vs. control. (**b**,**d**) Representative Western blots for TJ proteins and cleaved caspase-3, used for analysis of their content. For total protein used as loading control in Stain-Free Western blot analysis, see Appendix A.

## Data Availability

The data that support the findings of this study are available from the corresponding author upon reasonable request.

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
