# Peer review of "Dose- and Segment-Dependent Disturbance of Rat Gut by Ionizing Radiation: Impact of Tight Junction Proteins"

_ijms, 2023, doi:10.3390/ijms24021753_

Round 1
Reviewer 1 Report
This work is well designed and executed using sound methodology. I find the results to strongly support the conclusion drawn.
Author Response
Authors highly appreciate and strongly thank the reviewer for the pleasant comments of our work.

Reviewer 2 Report
Summary of the manuscript:
= = = = = = = = = = = = = = = = = = = = = = = = = = = = = = = = = = =
Livanova et al. explores the effects of IR-induced injury in male Wistar rats. They examined the jejunum and colon at 72 hours after exposure to 2 or 10Gy, the latter of which may produce acute gastrointestinal syndrome. In the jejunum irradiated with 10 Gy, they found changes in transepithelial resistance (TER), short circuit current (Isc), and paracellular permeability using sodium fluorescein. The colon had no observed change in permeability however TER and Isc were altered as in the jejunum. There were structural changes in both tissue sections as seen by histopathological H&E staining. In the jejunum, there were changes in expression by 10Gy for tight junction (TJ) complexes measured by western blots for Claudin-1, -2, -3, -4, tricellulin and occludin as well as measurements of cleaved Caspase-3. Note, only Claudin-2 and -4 were significantly altered by 10Gy in the colon. Proper approval for this study was made by St. Petersburg State University’s Ethical Committee for Animal Research (No. 131-03-5; December 13, 2017).
Specific comments for the authors:
= = = = = = = = = = = = = = = = = = = = = = = = = = = = = = = = = = =
1.) Abstract (line 17) “Severe violation(?) of histological structure” is an odd word choice. Please consider using disturbance, disruption, or other similar synonyms.
2.) Figure 2a-2d. It is unclear what the bar plots represent. Are the 5 minute time-course measurements (dynamics) a representative of each individual tissue measured (values)? Is there a specific time point selected across these samples displayed in the barplots?
3.) Line 83. The use of and asterisk (*) symbol is unconventional and should be replaced here and in other occurrences with a multiplication symbol.
4.) The authors previously published a parallel study in Male Wistar rats exposed to 2, 5 or 10Gy [PMID: 34337895; PMCID: PMC8326886] and examined identical endpoints in the jejunum (transepithelial resistance, short circuit current, paracellular permeability of sodium fluorescein, and histological examination). There is no mention of that study in the current manuscript which justifies the use of the 72 hour time-point as well as a comparison between 2Gy and 10Gy which are above and below the established LD50/30 in this animal model.
5.) Figure 3. There are 12 data points in each bar plot for control, 2Gy, and 10Gy samples which is described as “The number of symbols corresponds to the number of H&E sections analyzed” in the figure legend. It would be appropriate to specify the number of animals that these data points came from as a clearer indication of the statical significance.
6.) Line 124. “less significant impairment of TER and Isc of colon”. The interpretation of this statement is unclear. Comparisons between the jejunum and colon have the same statistical significance in the bar plots. Is this statement reflective of the relative differences in the absolute TER or Isc measurements or relative fold-changes to the control?
7.) Consider changing the term “reorganization” which used when describing the observed changes in TJ measurements (Lines 13, 195, 262). Changes in protein expression levels were directly measured here. Reorganization implies a change in physical cellular location, which would require confirmation with imaging via histology or immunofluorescence).
8.) Figure 4b and 4d. Please label the representative western blot columns with the irradiation condition.
9.) Figure 4b. (Assuming an order of 0, 2, 10Gy), the representative image chosen for Claudin-3 shows in marked decrease from the 2Gy sample, although slightly increased relative to the 0Gy control. It appears one outlier sample is responsible for increasing the mean. Perhaps a more representative image is appropriate to reflect the relative relationship between 2 and 10Gy for this protein.
10.) Lines 215-217. The cited decrease of Claudin-3 occurred in the terminal ileum at day 6 following 10Gy in Wistar rats [REF #20], which did show protein expression of Claudin-3 in the control 0Gy samples in that tissue. The lack of expression in the jejunum for Claudin-3 here is also supported by REF #17 and the observed increase does not necessarily support the statement “This may reflect an increase in claudin-3 occurred in the early stages of IR response, which declines as radiation injury progresses.”
11.) In the discussion (line 266-267), TJ proteins are implicated as countermeasure for GI-ARS. However, studies have shown mitigation of ROS via free radical scavengers such as N-acetyl-cysteine (NAC) have shown significant protection as well as mitigation from GI-ARS. A brief discussion of this mechanism was made on page 7 (starting at line 237). Are there any drug candidates in the literature that could be explored in the context of TJ proteins?
Author Response
Point 1: Abstract (line 17) “Severe violation(?) of histological structure” is an odd word choice. Please consider using disturbance, disruption, or other similar synonyms.
Response 1:This sentence has been rewritten. “These changes were accompanied by severe disturbance of histological structure…’ (line 17)
Point 2: Figure 2a-2d. It is unclear what the bar plots represent. Are the 5 minute time-course measurements (dynamics) a representative of each individual tissue measured (values)? Is there a specific time point selected across these samples displayed in the barplots?
Response 2: Bar plots demonstrate measurements of TER/ISc by the 5th min of registration in Ussing chamber, being a specific time point selected across dynamics data. Each symbol on the bar plot corresponds to a segment of tissue mounted in the Ussing chamber. The advantage of a bar plot over a line chart of TER/ISc in dynamics here is that it shows individual values ​​and their scatter, which is impossible to show on a trend plot.
The choice of just such a time point is because 5 min is a minor interval so that we can consider the registered measurements to be consistent with those that we could observe inside the body.
Figure 2 caption has been rewritten in order to proper describe the bar plots: (lines 103-108)
Point 3: Line 83. The use of and asterisk (*) symbol is unconventional and should be replaced here and in other occurrences with a multiplication symbol.
Response 3: Asterisk symbol has been replaced here (line 83) and in other occurrences (lines 345, 356, Figure 2a,b) with a multiplication symbol.
Point 4: The authors previously published a parallel study in Male Wistar rats exposed to 2, 5 or 10Gy [PMID: 34337895; PMCID: PMC8326886] and examined identical endpoints in the jejunum (transepithelial resistance, short circuit current, paracellular permeability of sodium fluorescein, and histological examination). There is no mention of that study in the current manuscript which justifies the use of the 72 hour time-point as well as a comparison between 2Gy and 10Gy which are above and below the established LD50/30 in this animal model.
Response 4: The following text has been added to discussion (lines 171-175):
Previously we examined TER, Isc, paracellular permeability of sodium fluorescein, and histological structure of jejunum under 2, 5 or 10Gy exposure, considering 72 h as an optimal interval to assess irradiation effects in Wistar rats [27]. Here we use 72 h post irradiation time point and compare 2 Gy and 10 Gy doses, which are above and below the established LD50/30 in this animal model [37].
The following reference was added:
- Challapalli, S.; Kumar, A.; Rai, R.; Kini, J.; Kumarchandra, R. Standardization of mean lethal dose (LD 50/30) of X-rays using linear accelerator (LINAC) in Albino Wistar rat model based on survival analysis studies and hematological pa-rameters. Res. J. Pharm. Biol. Chem. Sci. 2015, 5, 1215–1219.
Point 5: Figure 3. There are 12 data points in each bar plot for control, 2Gy, and 10Gy samples which is described as “The number of symbols corresponds to the number of H&E sections analyzed” in the figure legend. It would be appropriate to specify the number of animals that these data points came from as a clearer indication of the statical significance.
Response 5: Detailed description of the number of animals, tissue fragments and slides analyzed is provided in Materials and Method section (lines 361-362, 372-376):
Segments of the jejunum and large intestine obtained from sham-irradiated animals (n=3), as well as animals irradiated at a dose of 2 Gy (n=3) and 10 Gy (n=3)… In each animal, two tissue fragments were obtained, for each of which there were at least two H&E stained sections; at least five visual fields were examined per each section. Data on every five visual fields averaged per each H&E stained section was taken as an independent event. Independent double-blind analysis was conducted while providing quantitative assessment of tissue histological structure.
To sum up, each symbol in Fig.3 bar plot corresponds to the number of H&E sections analyzed, with 5 visual fields examined and averaged per each section. To clarify this procedure we added a mention of Materials and Methods section in Figure 3 caption (line 118) and specified the procedure in (lines 374-375).
Point 6: Line 124. “less significant impairment of TER and Isc of colon”. The interpretation of this statement is unclear. Comparisons between the jejunum and colon have the same statistical significance in the bar plots. Is this statement reflective of the relative differences in the absolute TER or Isc measurements or relative fold-changes to the control?
Response 6: Here we speculate about relative fold-changes to the control. For instance, control TER in jejunum was 76±5 Ohm*cm2 and in 10-Gy group it was 16 ± 2 Ohm*cm2, which means 79% decrease. At the same time in colon control TER value was 66 ± 3 Ohm*cm2, with 42 ± 2 Ohm*cm2 in 10-Gy group (36% decrease). However, the same observation of ISc (94% vs. 111%) does not actually demonstrate more dramatic alteration in jejunum, thereby the sentence was rewritten and clarified to be more precise (lines 125-127):
At the same time, TER in jejunum was altered more dramatically comparing to colon (79% decrease comparing to control in jejunum and 36% in colon). Less significant impairment of TER in colon, as well as unchanged permeability indicates increased colonic barrier radioresistance compared to jejunum, confirmed earlier [30,31].
Point 7: Consider changing the term “reorganization” which used when describing the observed changes in TJ measurements (Lines 13, 195, 262). Changes in protein expression levels were directly measured here. Reorganization implies a change in physical cellular location, which would require confirmation with imaging via histology or immunofluorescence).
Response 7: Given multidirectional changes in claudins, occludin and tricellulin, we consider dramatic shifts in TJ composition. At the same time we agree that we cannot speculate about dimensional reorganization of TJs. Term “reorganization of TJ” was replaced with “rearrangement of TJ content”, or “alteration of TJ”, or “disturbance” in all occurrences (lines 13, 18, 199, 203, 285)
Point 8: Figure 4b and 4d. Please label the representative western blot columns with the irradiation condition.
Response 8: Western blot columns were labeled with the irradiation condition (Fig. 4b, 4d)
Point 9: Figure 4b. (Assuming an order of 0, 2, 10Gy), the representative image chosen for Claudin-3 shows in marked decrease from the 2Gy sample, although slightly increased relative to the 0Gy control. It appears one outlier sample is responsible for increasing the mean. Perhaps a more representative image is appropriate to reflect the relative relationship between 2 and 10Gy for this protein.
Response 9: In case of Claudin-3 there was always (n=4) no band in control group, with intense band in 2 Gy group and another intense band for 10-Gy group, although slightly decreased compared to 2 Gy (having one outlier with increased intensity). Anyway, there was no statistically strong difference between 2 Gy and 10 Gy groups.
The image of WB bands for Claudin-3 was replaced with more appropriate to reflect the most common ratio between 2 Gy and 10 Gy for this protein (Fig. 4b)
Point 10: Lines 215-217. The cited decrease of Claudin-3 occurred in the terminal ileum at day 6 following 10Gy in Wistar rats [REF #20], which did show protein expression of Claudin-3 in the control 0Gy samples in that tissue. The lack of expression in the jejunum for Claudin-3 here is also supported by REF #17 and the observed increase does not necessarily support the statement “This may reflect an increase in claudin-3 occurred in the early stages of IR response, which declines as radiation injury progresses.”
Response 10: The authors agree with the reviewer's opinion regarding the inaccuracy of this speculation. The sentence has been replaced with the following statement (lines 223-225):
This can be explained by the fact that claudin-3 is not detected in the intact jejunum of Wistar rats, and the mechanisms of its participation in the radiation reaction can be different compared to another gut segment investigated.
Point 11: In the discussion (line 266-267), TJ proteins are implicated as countermeasure for GI-ARS. However, studies have shown mitigation of ROS via free radical scavengers such as N-acetyl-cysteine (NAC) have shown significant protection as well as mitigation from GI-ARS. A brief discussion of this mechanism was made on page 7 (starting at line 237). Are there any drug candidates in the literature that could be explored in the context of TJ proteins?
Response 11:The following text was included into discussion (lines 268-281):
Various approaches are implicated to reduce damaging effects of IR as countermeasures for GI-ARS. Mitigation of ROS via free radical scavengers such as N-acetyl-cysteine (NAC) have shown improvement in malondialdehyde and glutathione levels, as well as caspase-3 expression in small intestine of rats [10]. Some studies highlight the role of TJ proteins in gut protection with drug candidates. Neurotensin, gut tride-capeptide that stimulates growth of normal gut mucosa and pancreas, was shown to sig-nificantly reduce bacterial translocation and restore the structure of the villi caused by IR exposure, with subsequent influence on claudin-3 level in mice [20]. In our previous study, chronic ouabain administration was proved to modulate the transport and barrier functions of rat colon epithelium damaged by IR exposure [60]. There is growing evi-dence that oubain in nanomolar concentrations can affect the expression of claudins and epithelium barrier properties by triggering cSrc/Erk1/2 intracellular signal pathways and Na,K-ATPase [29, 61] which makes it a potential candidate to protect gut from ionizing radiation in the context of TJ proteins.
The following references were added:
- Kravtsova, V.V.; Fedorova, A.A.; Tishkova, M.V.; Livanova, A.A.; Vetrovoy, O.V.; Markov, A.G., Markov, A.G.; Matchkov, V.V.; Krivoi, I. I. Chronic Ouabain Prevents Radiation-Induced Reduction in the α2 Na, K-ATPase Function in the Rat Diaphragm Muscle. Int. J. Mol. Sci. 2022, 23, 10921. doi: 10.3390/ijms231810921.
- Larre, I.; Lazaro, A.; Contreras, R.G.; Balda, M.S.; Matter, K.; Flores-Maldonado, C.; Ponce, A.; Flores-Benitez, D.; Rin-con-Heredia, R.; Padilla-Benavides, T.; Castillo, A.; Shoshani, L.; Cereijido, M. Ouabain modulates epithelial cell tight junction. Proc. Natl. Acad. Sci. U. S. A. 2010, 107, 11387-11392. doi: 10.1073/pnas.1000500107.
Reviewer 3 Report
High doses irradiation causes severe intestinal dysfunction that characterizes the gastrointestinal acute radiation syndrome (GIARS). The authors addressed the dose effect and segment-specificity of TJ proteins impact in IR-induced gastrointestinal injury in rats. The topic is very interesting, novel and important; however, the authors only have the western blot data of tight junction proteins, we don't know the exact structure change or mechanism of tight junction change unless we have the immune histochemistry staining or immune fluorescence staining of the tight junction proteins.
Some specific concerns,
1) For Figure 1b, do you have the blood count before the radiation, or the blood count for the 2 Gy group?
2) For Figure 2, how many technical repeats and biological repeats you have done. I assume you have 10 control mice, 4 X 2 Gy mice, and 6 X 10 Gy mice, the biological repeat will be your animal number. However, the dots in your bar graph of Figure 2 are more than the animal number, can you specify how many repeats you have used for Figure 2. I know if you have more repeats (n) , and you will lower your standard error, and you will have more chance to see the difference. But you need to mention it in your figure legend. The same for Figure 3.
3) For Figure 3, because you claimed that irradiation change the structure of intestine and colon, can you put your high resolution HE image, so we can see the structures?
4) For Figure 4, we need loading control for the western blot, either by total protein loading or by staining a housekeeping gene.
Round 2
Reviewer 3 Report
Happy new year and great works!